

# Bite force estimates in juvenile *Tyrannosaurus rex* based on simulated puncture marks

Joseph E. Peterson[1], Z. Jack Tseng[2] and Shannon Brink[3]

[1] Department of Geology, University of Wisconsin Oshkosh, Oshkosh, Wisconsin, United States of America
[2] Department of Integrative Biology and Museum of Paleontology, University of California Berkeley, Berkeley, California, United States of America
[3] Department of Geological Sciences, East Carolina University, Greenville, North Carolina, United States of America

Corresponding author
Joseph E. Peterson,
petersoj@uwosh.edu

## ABSTRACT

**Background:** Bite marks attributed to adult *Tyrannosaurus rex* have been subject to numerous studies. However, few bite marks attributed to *T. rex* have been traced to juveniles, leaving considerable gaps in understanding ontogenetic changes in bite mechanics and force, and the paleoecological role of juvenile tyrannosaurs in the late Cretaceous.

**Methods:** Here we present bite force estimates for a juvenile *Tyrannosaurus rex* based on mechanical tests designed to replicate bite marks previously attributed to a *T. rex* of approximately 13 years old. A maxillary tooth of the juvenile *Tyrannosaurus* specimen BMR P2002.4.1 was digitized, replicated in dental grade cobalt chromium alloy, and mounted to an electromechanical testing system. The tooth was then pressed into bovine long bones in various locations with differing cortical bone thicknesses at varying speeds for a total of 17 trials. Forces required to replicate punctures were recorded and puncture dimensions were measured.

**Results:** Our experimentally derived linear models suggest bite forces up to 5,641.19 N from cortical bone thickness estimated from puncture marks on an *Edmontosaurus* and a juvenile *Tyrannosaurus*. These findings are slightly higher than previously estimated bite forces for a juvenile *Tyrannosaurus rex* of approximately the same size as BMR P2002.4.1 but fall within the expected range when compared to estimates of adult *T. rex*.

**Discussion:** The results of this study offer further insight into the role of juvenile tyrannosaurs in late Cretaceous ecosystems. Furthermore, we discuss the implications for feeding mechanisms, feeding behaviors, and ontogenetic niche partitioning.

## INTRODUCTION

Bite mechanics and feeding habits of dinosaurs have long been debated. A variety of methods have been proposed to determine bite mechanics and bite forces of members of Dinosauria, including stegosaurs, ceratopsians and hadrosaurids (*Weishampel, 1984*; *Bell,*

*Snively & Shychoski, 2009*; *Reichel, 2010*; *Erickson et al., 1996b*), and more commonly, theropods (*Rayfield et al., 2001*; *Rayfield, 2005*; *Rayfield et al., 2007*; *Gignac et al., 2010*; *Lautenschlager et al., 2013*). The genus *Tyrannosaurus rex* and other tyrannosaurids have been the focus of many studies on dinosaur bite force and bite mechanics (*Erickson et al., 1996a*; *Meers, 2002*; *Barrett & Rayfield, 2006*; *Bates & Falkingham, 2012*; *Gignac & Erickson, 2017*; *Rowe & Snively, 2021*; *Therrien et al., 2021*). These studies have relied on several methods for estimating bite forces, including multi-body dynamic analysis (MDA) (*Bates & Falkingham, 2012*), finite element analysis (*Rayfield, 2005*; *Rayfield et al., 2007*; *Maiorino et al., 2015*), and actualistic studies.

However, bite force estimates have largely focused on adult specimens with few studies providing estimates for juveniles or subadult *T. rex*, leaving a considerable gap in the understanding of tyrannosaur ontogenetic dietary partitioning and paleoecology. *Bates & Falkingham (2012)* based their bite force estimate of a late-stage juvenile *T. rex* on MDA, suggesting allometric growth in bite force from juvenile to adult. The juvenile specimen used in that study (BMR P2002.4.1) was also found to possess bite marks through the left maxilla and nasal. These have been interpreted as conspecific bites by *Peterson et al. (2009)* based on the strong correlation between the dimensions and spacing of the punctures and the dentition of BMR P2002.4.1 itself (Figs. 1A–1E). Similarly, *Peterson & Daus (2019)* identified feeding traces on a proximal caudal vertebra from an *Edmontosaurus* (BMR P2007.4.1) likely produced by a *T. rex* of a similar ontogenetic stage using similar deductive methods to *Peterson et al. (2009)* (Figs. 2A–2E).

The presence of two sets of puncture marks attributable to a late-stage juvenile *T. rex* provides the opportunity to test previously derived juvenile *T. rex* bite force estimates from multi-body dynamic analyses (*Bates & Falkingham, 2012*) with actualistic methods (*Gignac et al., 2010*). Comparisons between the bite forces of adult and juvenile *T. rex* have the potential to reveal ontogenetic niche partitioning (*Woodward et al., 2020*) and illuminate the impact of *T. rex* ontogeny in terrestrial Cretaceous ecosystems.

## MATERIALS & METHODS

*Gignac et al. (2010)* reported on bite marks in a specimen of *Tenontosaurus tilletti* that were attributed to *Deinonychus antirrhopus*, and designed indentation experiments to determine bite force estimates for *D. antirrhopus*. We applied similar methods to estimate the bite force for a juvenile *Tyrannosaurus*. Previous studies of BMR P2002.4.1 ("Jane") and BMR P2007.4.1 ("Constantine") suggest that their respective bite marks were produced by a lateral maxillary tooth of a juvenile to sub-adult tyrannosaur (*Peterson et al., 2009*; *Peterson & Daus, 2019*). The trace on BMR P2002.4.1 penetrates through 7.5 mm of cortical bone, while the traces on BMR P2007.4.1 penetrates through 0.4 mm of cortical bone. Both sets of puncture marks are approximately 10–19 mm in length, and 4–9 mm in width (*Peterson et al., 2009*; *Peterson & Daus, 2019*). To replicate these indentations, a lateral maxillary tooth of the juvenile *Tyrannosaurus* specimen BMR P2002.4.1 was digitized and 3D printed. Triangulated laser texture scans were conducted at the Department of Geology at the University of Wisconsin–Oshkosh in Oshkosh, WI. Scans were made with a NextEngine 3D Laser Scanner, capturing data at seven scanning

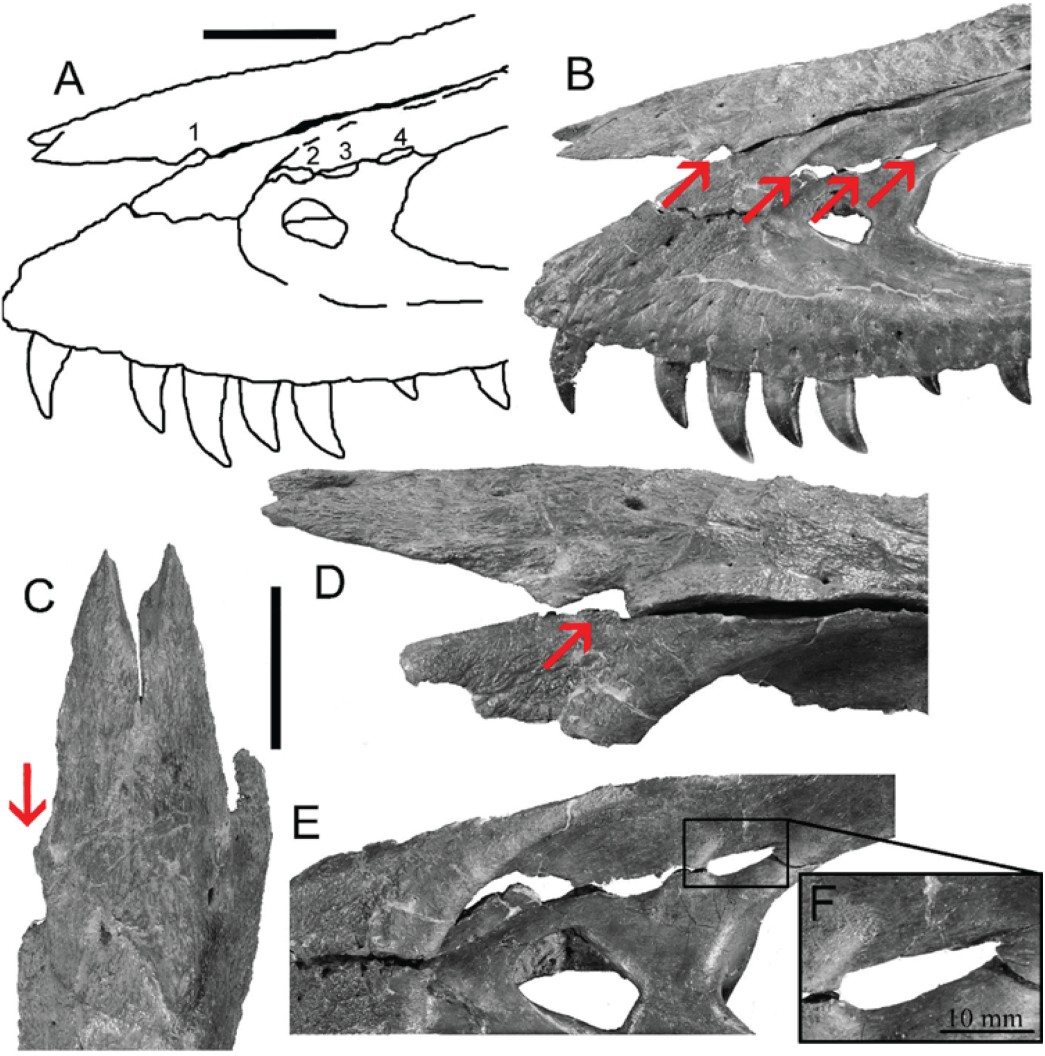

**Figure 1 Lesions present on the face of BMR P2002.4.1.** (A) A line drawing of the lesions (1–4). (B) Red arrows indicate the locations of the four lesions on the left maxilla and nasal of BMR P2002.4.1. (C) A dorsal view of the anterior nasal, with the red arrow indicating asymmetry resulting from the puncture on the left side. (D) The first puncture (arrow 5 lesion 1) located on the articular surface of the anterior nasal and left maxilla. (E) The three lesions on the left maxilla of BMR P2002.4.1 with a close-up of lesion 4 (F). Scale bars: A–B 5 10 cm; C–E 5 5 cm. Figure modified from *Peterson et al., 2009*.

divisions in high definition (2.0 k points/in$^2$). Models were built with the NextEngine ScanStudio HD Pro version 2.02 and finalized as an STL (stereolithograph) model (Fig. 3A). The STL file was then imported into Meshmixer (Autodesk, version 10.0.297), in which the 'Make Solid' algorithm was utilized to prepare the model for printing by filling 'gaps' in the model mesh as well as the removal of artifacts from the scanning process (*Peterson & Krippner, 2019*). While the digitization process produced a tooth model of the same dimensions as the original specimen, fine details such as denticles were lost in the digital processing. The digital model of the tooth was then fused to a model of an adapter that allowed the 3D printed model to be mounted onto the test frame (see below)

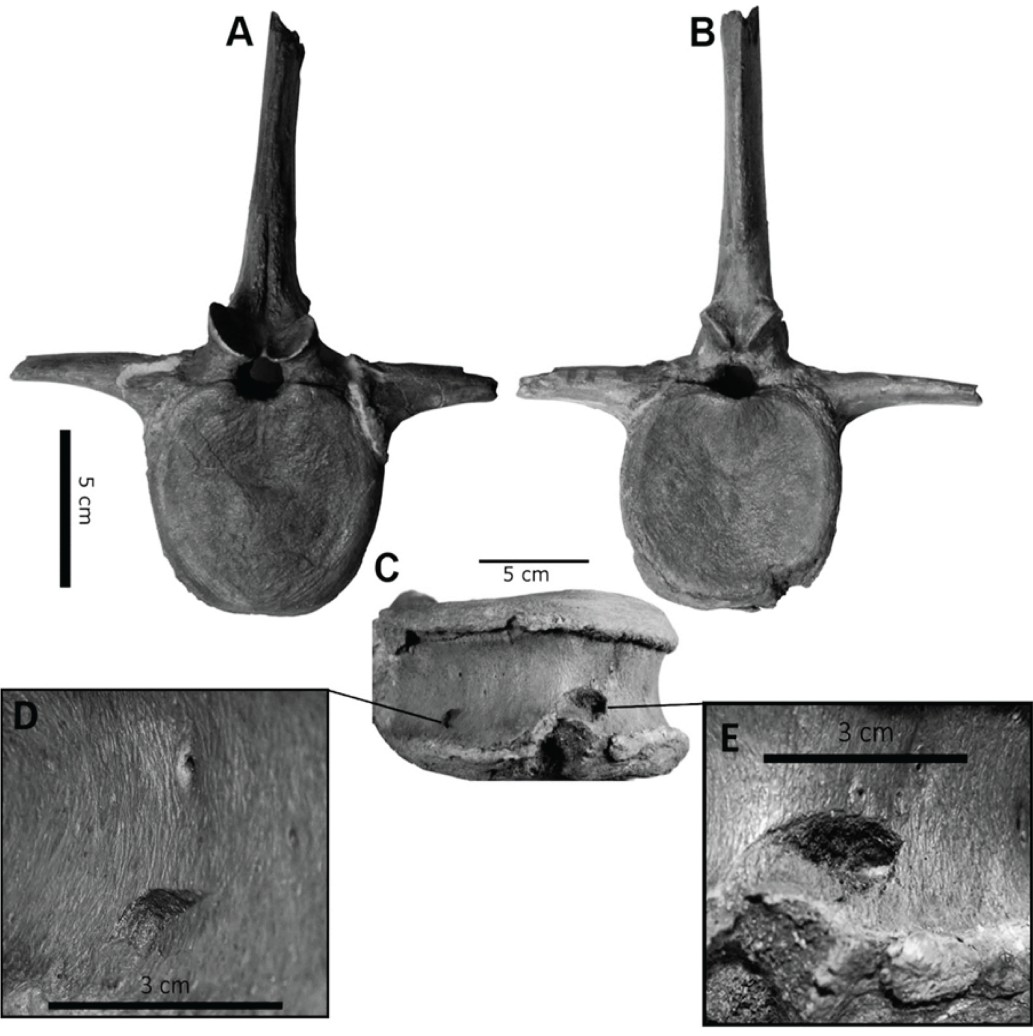

**Figure 2 Punctured caudal vertebra of BMR P2007.4.1.** Punctured caudal vertebra of BMR P2007.4.1. BMR P2007.4.1 in (A) anterior, (B) posterior, and (C) ventral views, including (D, E) the two elliptical punctures on the ventral surface of the centrum. Modified from *Peterson & Daus, 2019*.

using Geomagic Wrap (3D Systems, Cary, NC, USA). In order to produce a suitable tooth analog, the STL file was 3D printed in a dental grade cobalt chromium alloy [Co (61.0), Cr (25.0),Mo (6.0),W (5.0),Mn (<1.0),Si (<1.0), Fe (<1.0)] with a yield strength of 47,436 N/cm$^2$ (474.36 MPa) (Fig. 3B) by the Argen corporation (San Diego, CA, USA) to serve as a rigid model relative to the testing medium (i.e., cortical bone). The compressive strength of the alloy model was higher than that of tooth enamel (384 MPa) and dentin (297 MPa) (*Willems et al., 1993*). However, the differences in physical properties between the alloy tooth model and tyrannosaur teeth were irrelevant for the purposes of this study since the fossil bite marks suggest that tyrannosaur teeth were capable of producing bite marks in bone. In order to ensure that the alloy model was capable of withstanding similar stresses, a dental grade cobalt chromium alloy was chosen due to its high compressive strength and rigidity that would be needed for the high vertical and

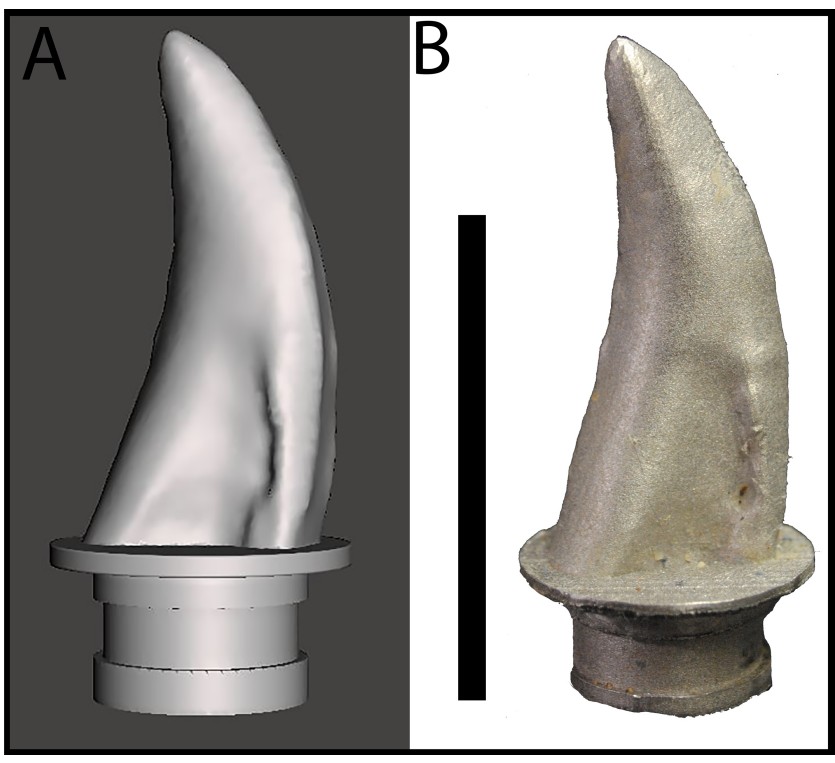

**Figure 3 Maxillary tooth model of BMR P2002.4.1.** (A) digital model, and (B) 3D print in cobalt used for bite force simulation experiments.

compressive loading the model would endure during testing (*Sharir, Barak & Shahar, 2008*).

The dental grade cobalt chromium alloy tooth model was mounted to a Shimadzu AGS-X Universal Electromechanical Test Frame (ETF) equipped with a 10 kN load cell, interfaced with the Shimadzu TrapeziumX software for data collection. Prior experiments on bite force determination have utilized bovine limb bones for their varying cortical thicknesses and similarity in microstructure to dinosaurian elements (*Erickson, Catanese & Keaveny, 2002*; *Locke, 2004*; *Gignac et al., 2010*). While the elements under study include cranial and vertebral elements that may differ in microstructure than limb elements, the comparable variance in cortical thickness makes bovid limb elements suitable models for these experiments.

A fresh right bovine humerus and an in-tact left radius/ulna pair, sourced from a local meat market were used for indent simulations. The bones had muscle and other soft tissue removed, were kept frozen upon purchase and thawed overnight at room temperature before testing proceeded. Epiphyses were left intact to reduce the possibility of creating microfractures and compromising structural integrity during removal. Bones were covered with paper towels soaked in Hank's Balanced Salt Solution (HBSS) between tests to maintain moisture. The bones were secured to the lower stage of the ETF using a series of one-inch width nylon straps. Testing parameters including maximum allowed force (10 kN), maximum displacement (ranging from 5–45 mm depending on depth of the test

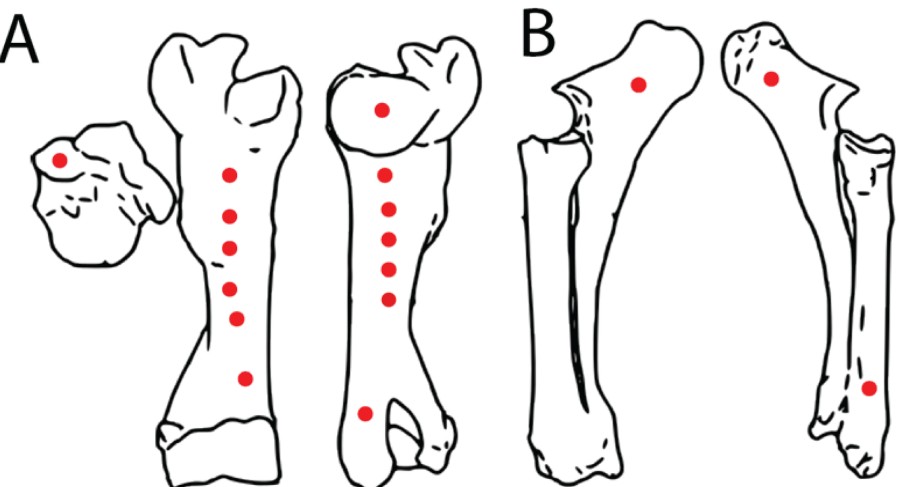

**Figure 4** **Spatial maps showing experimental indentation locations on bovine long bones.** (A) Right humerus, (B) left radius and ulna.

location on specimen), and speed (1 mm/s) were set. Bone mechanical behavior is loading rate-dependent, with increasing strength at higher loading rates (*McElhaney, 1966*). We adopted a conservative load rate of 1 mm/s following *Gignac et al. (2010)*, and additionally conducted a limited number of trials at the higher but physiologically more realistic rate of 10 mm/s ($n = 3$ trials) (*Erickson et al., 1996a*) as well as the machine-defined limit of 16 mm/s ($n = 2$ trials) to assess the sensitivity of the estimated puncture forces as a factor of load rate. The tooth model was then pressed into the bones in various locations with differing cortical bone thicknesses to produce a total of 17 indents (Figs. 4A, 4B). After each individual test, the resulting indent was measured using Mitutoyo vernier calipers for depth, width, and length to the nearest 0.02 mm before proceeding. We plotted load-displacement relationships of all trials runs and kept only trials with smooth curves as in *Gignac et al. (2010)*; curves that exhibited sudden drops in measured load indicate presence of fractures at and around the indentation site, and those curves were excluded from subsequent analyses. Furthermore, trials that exhibited any visible movements or shifts in the test specimen were excluded from subsequent analysis. During multiple trials that occurred in close spatial proximity to each other on the test bone specimens, we visually inspected the targeted puncture sites to make sure there are no visible cracks on the surface before conducting each trial. Post-indent testing, all specimens were scanned at 0.6 mm slice thickness using a GE Discovery 690 PET-CT scanner in the University at Buffalo Clinical and Translational Science Institute Image Center, Buffalo, NY, USA (Fig. 5).

To estimate the indentation forces required to make the specific puncture marks observed on BMR P2007.4.1 and BMR P2002.4.1, we used linear regression to model the relationship between per trial maximum recorded indentation force and puncture site cortical bone thickness, as in *Gignac et al. (2010)*. We then used the linear regression model to calculate the indentation forces required to make puncture marks with the measured cortical thickness values from BMR P2007.4.1 (0.4 mm) and BMR P2002.4.1 (7.5 mm).
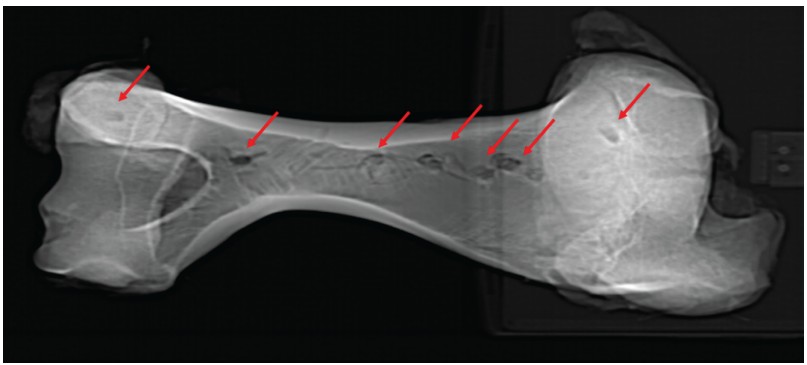

**Figure 5 Computed tomographic image of bovine right humerus post-indentation.** Computed tomographic image of bovine right humerus post-indentation. Red arrows indicate location of experimental indentations.

Additionally, we estimated the uncertainty around the calculated indentation forces using 95% confidence intervals around the linear regression model equation. These calculations were conducted in the R programming environment using the core functions *lm* and *predict*.

## RESULTS

The trial data were analyzed using linear modeling of bovine specimen cortical thickness and indentation force values and derived predictive formulae (1) with the full data set (force = 637.41 * cortical-thickness + 860.62; $R^2$ = 0.5361) and (2) with fractured trial values excluded (force = 628.48 * cortical-thickness + 555.72; $R^2$ = 0.6346) (Figs. 6A–6B). The full dataset predicts a force of 1,115.58 N for the indentation on BMR P2007.4.1 and 5,641.19 N for the indentations on BMR P2002.4.1; the fracture-excluded dataset predicts a force of 807.11 N for the indentation on BMR P2007.4.1 and 5,269.31 N on BMR P2002.4.1 (Table 1).

The resulting relationships between puncture force and cortical thickness at puncture site in the additional trials at higher load speeds are consistent with those obtained from the 1 mm/s trials. At 10 mm/s, cortical thicknesses ranging from 3.7 to 5.3 mm required forces of 2,930.4 to 10,448.6 N. At 16 mm/s, a cortical thickness of 4.5 mm correlated with a puncture force of 3,248.8 N, and a thickness of 9 mm correlated with 9,024.41 N.

The maximum force recorded by the 10 kN load cell was 10,448.60 N, and minimum 782.93 N. Summary of each trial and raw force-displacement time series data are available as Supplemental Data S1. A video of one of the experimental trials (S2) and CT images of the experimentally punctured cow elements (S3) are available on MorphoSource (https://www.morphosource.org/Detail/ProjectDetail/Show/project_id/1117).

## DISCUSSION

Estimated bite forces of adult *T. rex* have yielded a wide range of results, and our study provides the first experimentally derived juvenile bite force estimates to contextualize the assessment of adult bite force estimates. Modelled muscle volume estimates for adult

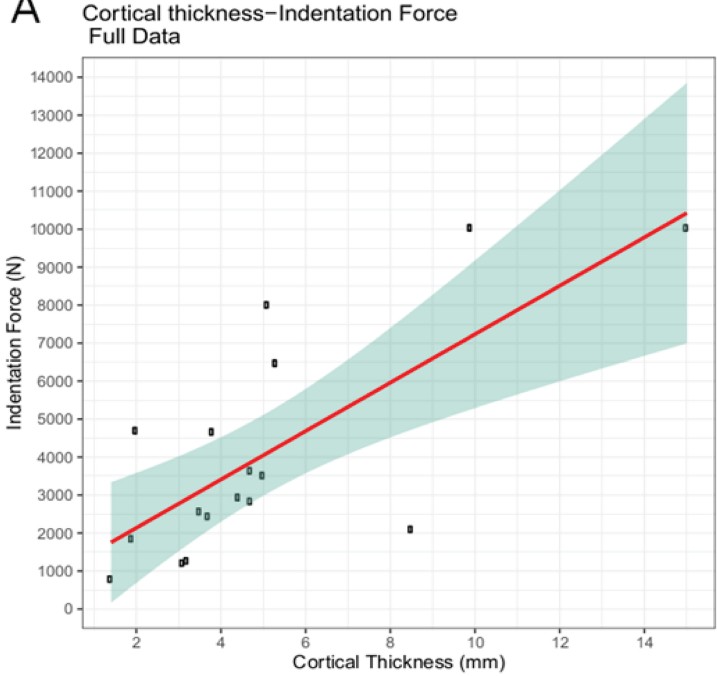

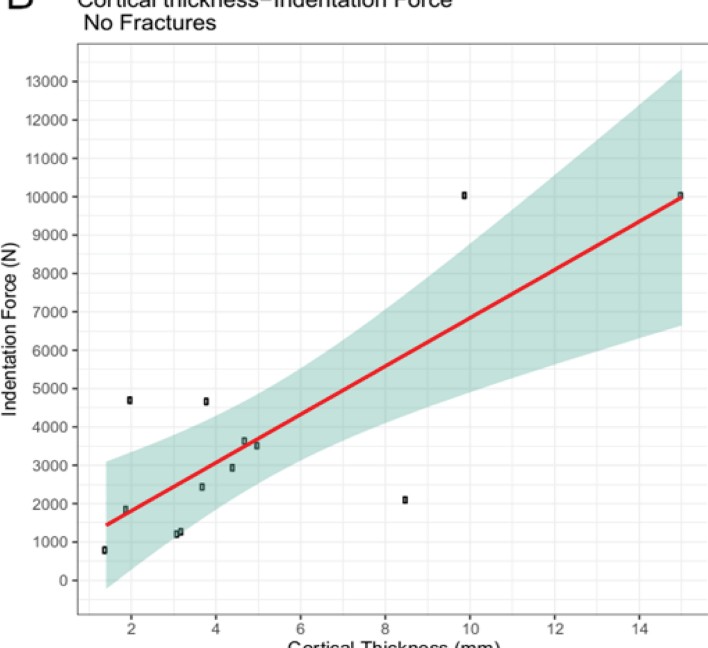

**Figure 6 Indentation force-cortical thickness plots for experimental data.** Indentation force-cortical thickness plots for experimental data. Thick line represents the fitted linear regression line. Shaded region represents 95% confidence intervals. (A) Analysis done using the full dataset that excluded trials with visible specimen movements (force = 637.41 * cortical-thickness + 860.62; $R_2$ = 0.5361); (B) Analysis excluding both indentation forces at or over the 10,000 N threshold, and those trials that showed evidence of fracture (force = 628.48 * cortical-thickness + 555.72; $R_2$ = 0.6346).

**Table 1 Puncture dimensions (mm), cortical thickness (mm), and measured force (N) of the 17 indentation trials.**

| Puncture dimensions (mm) | | | Cortical thickness | Force | Fracture |
|---|---|---|---|---|---|
| Length | Width | Depth | (mm) | (*N*) | |
| 11 | 7.7 | 10 | 1.4 | 782.929 | No |
| 7.3 | 5.7 | 10 | 1.9 | 1,844.92 | No |
| 9.5 | 8 | 10 | 2 | 4,690.79 | No |
| 10 | 8.1 | 10 | 3.1 | 1,202.62 | No |
| 6.9 | 5.2 | 6.8 | 3.2 | 1,263.62 | No |
| 6.7 | 4.1 | 1 | 3.5 | 2,562.17 | Yes |
| 8.8 | 5.2 | 10 | 3.7 | 2,432.3 | No |
| 10.4 | 5.4 | 10 | 3.8 | 4,657.54 | No |
| 2.8 | 3.7 | 4 | 4.42 | 2,931 | No |
| 8.3 | 5.7 | 10 | 4.7 | 2,830.95 | Yes |
| 8.4 | 5 | 10 | 4.7 | 3,630.72 | No |
| 9.3 | 5.3 | 10 | 5 | 3,509 | No |
| 8.8 | 5.3 | 10 | 5.1 | 8,000 | Yes |
| 8.5 | 5.6 | 10 | 5.3 | 6,463.85 | Yes |
| 11.9 | 3.8 | 4 | 8.5 | 2,094.83 | No |
| 6 | 3.4 | 8.5 | 9.9 | 1,0028.7 | No |
| 5 | 5.6 | 7.9 | 15 | 1,0024.6 | No |

*T. rex* bite correspond to forces between 8,526 and 34,522 N (*Barrett & Rayfield, 2006*; *Bates & Falkingham, 2012*). However, estimates incorporating likely muscle fiber length produced results over 64,000 N for adult *T. rex* (*Bates & Falkingham, 2018*). Furthermore, the unique tooth morphology and arrangement in adult *T. rex* promote fine fragmentation of bone during osteophagy (*Gignac & Erickson, 2017*). Juvenile *T. rex* such as BMR P2002.4.1 have much narrower and blade-like tooth morphologies (*Carr, 2020*) and were unlikely to have been able to withstand similar bite forces at this ontogenetic stage. *Bates & Falkingham (2012)* estimated a maximum bite force for BMR P2002.4.1 at 2,400–3,850 N and hypothesized that ontogenetic increases in bite force could indicate a change in dietary partitioning and feeding behavior while approaching adulthood. Our experimentally derived linear models suggest bite forces of 5,269.31 to 5,641.19 N from cortical bone thickness estimated from puncture marks on a juvenile *Tyrannosaurus* (BMR P2002.4.1). These results suggest indentation forces up to 235% of previous estimates for juveniles. However, the lack of serration denticles on the dental grade cobalt alloy tooth model used in this study may slightly influence these results (*Abler, 1992*).

The testing equipment used in this study has a limit of 10,000 N. However, most of the results were well below this limit, suggesting that mechanical limits of the equipment were not a factor in the results. Furthermore, the load cell on the test frame is rated for 10,000 N, with a built-in safety factor of ~5% over the listed limit. Therefore, it is possible that the values at and over 10,000 N may be truncated. To assess the effect of potential truncation bias on our linear model estimates of indentation force, we repeated

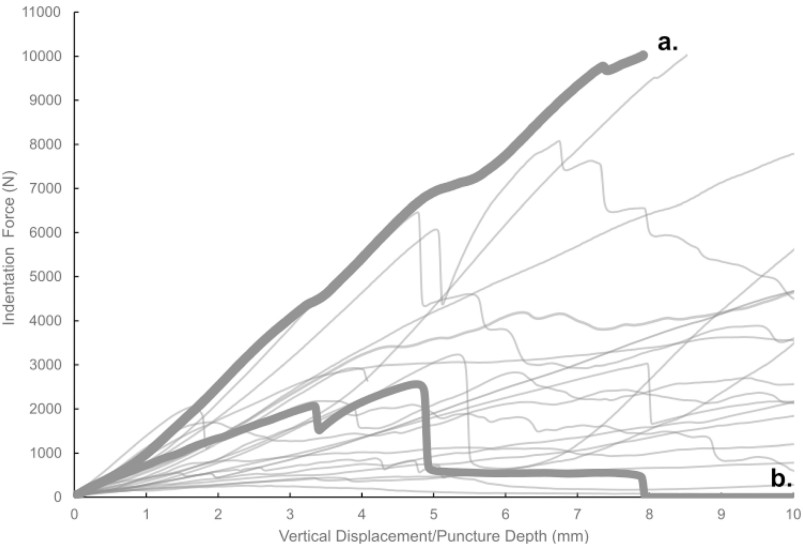

**Figure 7 Force-displacement curves from all experimental trials conducted in this study.** Force-displacement curves from all experimental trials conducted in this study. A typical smooth curve (A) exhibits no sudden drops, whereas curves indicative of fracture (B) show such drops.

the analysis by excluding force values at and over 10,000 N; the resulting indentation force estimates for the fossilized bite marks vary by 16–17% (higher in the "Constantine" specimen with trimmed data, lower in the "Jane" specimen with trimmed data). Conservative adjustment of all model-predicted indentation forces by a factor of 17% on both ends still returns values higher than previous estimates (4,373.53–4,682.19 N with 17% reduction vs. 2,400–3,850 N reported by Bates & Falkingham).

The range of material properties present (not quantified) throughout the test samples may be partially responsible for the variability in puncture forces measured at a given cortical thickness, and explain the similar results obtained in this study using different puncture rates (i.e., higher loading rates at less stiff locations may result in similar required puncture forces as low loading rates at stiffer locations on the bone sample).

Most of the force-displacement curves from experimental trials exhibit a stereotypical linear or near-linear initial portion, consistent with expectations from first principles of bone mechanics within the elastic region of a material force-displacement or stress-strain curve. In contrast, all but three of the force-displacement curves exhibited no clear peak force/stress; instead, the bone puncture continued to enlarge with additional penetration depth, with oscillating force magnitudes (Fig. 7) (*Erickson et al., 2004*; *Gignac et al., 2010*). While these results do not permit absolute determination of whether the bites studied were made with the animals' highest possible bite force, they do offer insight into the minimum boundary for the bite force capabilities of a late-stage juvenile *T. rex*.

We observed that the irregularly shaped epiphyses of the bovine bone specimens sometimes generated minor to substantial movements of the test specimen relative to the testing frame during bite trials, despite the use of nylon straps to secure the specimens.
Test trials that exhibited visible movements of the bone were removed from data analysis, but it is likely that minute movements took place during some of the bite force experiments. Consequently, we did not discuss the bite force trials individually, and we instead relied on regression model derived values as more robust estimates of the bite force values used in our linear model-based estimates of bite force, which were in turn based on cortical bone thickness at puncture mark sites of fossil specimens. The possible movement of bone specimens during a given bite experiment is not an unrealistic factor in the actual feeding and predatory behavior being studied, as movements of multiple bodies are involved in generating puncture marks from agonistic or hunting behavior in predators. Future studies that include a more formalized consideration of potential multibody dynamics of a particular bite would provide further refinement on such bite force estimates. We opted to maintain the unmodified state of bone specimens in our trials, rather than processing those samples into standardized shapes (e.g., cubes, cylinders), in order to minimize inadvertent damage to samples from cutting and to maximize the number of testing locations on each specimen. As such, the flexure of specimens is considered alongside flexures of the components of the testing frame itself as systematic errors in the study design that added variability to our measured values. Accordingly, the reported findings should be considered in this context.

Another important caveat to keep in mind is that multiple puncture trials were conducted in relatively close spatial proximity to each other on the test specimens (especially on the humerus specimen; Fig. 4A). Although we visually inspected each target puncture site prior to setting up each experimental trial to ensure there were no visible surface fractures, it is possible that internal and/or micro-fractures not visible to us were present at certain puncture sites. We diligently maintained specimen moisture during the experimental trials, thus minimizing the increase in brittleness of the bone from dehydration. Nevertheless, the results should be interpreted with this caveat in mind.

The tooth marks observed on BMR P2002.4.1 and BMR P2007.4.1 are Type 1 punctures (*Jacobsen, 1998*; *Tanke and Currie, 1998*), described as "*punctures (partial and full penetration) are circular to oval in outline. In unhealed examples, plates of bone are folded down and inwards into the puncture hole. The tooth/teeth are pushed into the bone and extracted with no additional damage*" (*Jacobsen, 1998*). *Erickson & Olson (1996)* note that the tooth marks most attributed to *T. rex* are classified as Types 1 and 2 ("*Transverse gouges, scores or tooth drag imprints are elongate, gently curving lesions with ragged (or healing) margins*" which are also known as "pull and puncture") (*Erickson et al., 2004*; *Carr, 2020*).

Similar Type 1 punctures have been observed on the skulls of fossil and extant crocodiles (*Buffetaut, 1983*; *Katsura, 2004*; *Peterson et al., 2009*) and attributed to intraspecific fighting. Intraspecific facial biting in crocodylians involves rapid single bites to the face with quick inertial movements (*Kalin, 1936*; *Webb & Messel, 1977*; *Webb, Manolis & Buckworth, 1983*; *Bramble & Wake, 1985*; *Lang, 1987*; *Cleuren & De Vree, 2000*; *Schwenk, 2000*; *Njau & Blumenschine, 2006*; *Hiiemae & Crompton, 2013*). Moderate to severe injuries from interspecific and intraspecific aggression have also been observed in

juvenile Cinereous Vultures (*Aegypius monachus*) and Griffon Vultures (*Gyps fluvus*) that were observed fighting over access to a carcass (*Blanco et al., 1997*).

Feeding behaviors in crocodylians involve similar sequences of biting and quick inertial movements (*Cleuren & De Vree, 2000*; *Njau & Blumenschine, 2006*; *Noto, Main & Drumheller, 2012*). However, during feeding carcasses are commonly dismembered through more vigorous inertial movements such as "death-rolling" where the predator will spin its body along the longitudinal axis while gripping the carcass in its jaws (*Schimdt, 1944*; *Attwell, 1958*; *Green, 1988*; *Njau & Blumenschine, 2006*). This method of carcass reduction permits inertial feeding and often produces abundant and deep-penetrating tooth marks that are morphologically similar to Types 1 and 2 tooth marks observed in theropod dinosaurs (*Njau & Blumenschine, 2006*).

Bite marks and feeding traces attributed to theropod dinosaurs have been extensively studied (*Fiorillo, 1991*; *Carpenter, 1998*; *Chure, Fiorillo & Jacobsen, 1998*; *Jacobsen, 1998*; *Tanke and Currie, 1998*; *Farlow & Holtz, 2002*; *Fowler & Sullivan, 2006*; *Happ, 2008*; *Bell & Currie, 2009*; *Peterson et al., 2009*; *Gignac et al., 2010*; *Peterson & Daus, 2019*; *Eberth & Currie, 2010*; *Hone and Rauhut, 2010*; *Hone & Watabe, 2010*; *Longrich et al., 2010*; *DePalma et al., 2013*; *Hone & Tanke, 2015*; *McLain et al., 2018*; *Drumheller et al., 2020*). Whereas most studies discuss tooth marks attributable to adult theropods, the ability to observe multiple bitten specimens from a juvenile offer further insight into potential ontogenetic shifts in diet and behavior (e.g., *Schroeder, Lyons & Smith, 2021*). Bite marks specifically attributable to intraspecific aggression in theropods include Types 1 and 2 tooth marks to the maxilla, nasal, jugal, and dentary (*Tanke and Currie, 1998*; *Bell & Currie, 2009*; *Peterson et al., 2009*). In crocodylians, these injuries are produced during rapid biting motions directed at the face, which is covered in relatively thin integument and tissues, requiring less resistance from muscle prior to striking bone.

Alternatively, traces from feeding can occur in a wide variety of skeletal locations, and the amount of soft tissue present at the time of the bite can have considerable ramifications for the morphology of the bite mark. For example, the bitten caudal vertebra of BMR P2007.4.1 is from the cranial-most part of the tail where substantial muscles such as the M. ilio-ischocaudalis and M. caudiofemoralis longus would have been present in life (*Snively & Russell, 2007*; *Peterson & Daus, 2019*). However, the punctures are present on the ventral surface of the centrum, suggesting that the tyrannosaur was feeding after the haemal complexes and most of the superficial hypaxial muscles and M. caudofemoralis longus had been removed (*Peterson & Daus, 2019*). Despite the inferred later-stage feeding, the punctures on BMR P2007.4.1 still penetrate approximately 5 mm in depth. Similar penetrating marks occur from crocodylians during the disarticulation of a carcass (i.e., "death-rolling"). The modelled craniocervical musculature of adult *T. rex* (based on analysis of superficial muscular reconstructions of the M. transversospinalis capitis, M. complexus, and *M. longissimus* capitis superficialis) suggest rapid strikes and inertial feeding similar to what is seen in extant archosaurs (*Snively & Russell, 2007*). Furthermore, *Snively et al. (2014)* confirmed these actions in birds, through observations of EMG and kinematics; chickens and eagles roll their heads to vigorously shake small prey or tear flesh, respectively. Considering the high leverage from the broad skulls of juvenile and

adult *T. rex*, such behavior would likely have been utilized for prey dismemberment analogous to crocodylian "death-rolling". As such, the feeding traces present on BMR P.2007.4.1 may have been the result of dismemberment of the carcass by a juvenile tyrannosaur.

These experimental reconstructions of the punctures present on BMR P2002.4.1 and BMR P2007.4.1 suggest that late-stage juvenile and subadult tyrannosaurs were capable of puncturing bone during feeding and intraspecific aggressive bouts despite the absence of the large, blunt dental crowns of adults (*Woodward et al., 2020*). The tooth marks present on BMR P2007.4.1 are consistent with feeding traces during dismemberment, possibly while a significant amount of soft tissue was still present (*Peterson & Daus, 2019*). Alternatively, the facial pathologies on BMR P2002.4.1 involved minimal tissue, thus a quicker, higher-inertia bite is likely, consistent with intraspecific aggression as seen in crocodylians (*Njau & Blumenschine, 2006*; *Peterson et al., 2009*). Further identification of tyrannosaur feeding traces from different ontogenetic stages may reveal more insight into the ecological role and potentially dynamic dietary trends of *T. rex* throughout ontogeny.

## ACKNOWLEDGEMENTS

The authors thank PeerJ Editor Andrew Farke, and reviewers Eric Snively, Lloyd Courtenay, and Stephanie Drumheller for their helpful comments and suggestions. We thank William W. Brink for laboratory assistance during experimentation trials, Jonathan P. Warnock, Christopher R. Noto, and M. Allison Stegner for stimulating discussion in the experimental design and constructive feedback on an early version of this manuscript. We also thank Moriarty Meats (1650 Elmwood Ave, Buffalo, NY, 14207, USA) for sourcing bone samples, and Paul Cascone and the Argen Corporation for assisting in the production of the dental grade cobalt chromium alloy tooth model used in this study.

### Funding

This work was supported by the UW Oshkosh Undergraduate Student/Faculty Research Collaborative Grant Program (2019–2020) and the National Institutes of Health (No. UL1TR001412). The funders had no role in study design, data collection and analysis, decision to publish, or preparation of the manuscript.

### Grant Disclosures

The following grant information was disclosed by the authors:
National Institutes of Health: UL1TR001412.

### Competing Interests

The authors declare that they have no competing interests.

Peer J

## Author Contributions

- Joseph E. Peterson conceived and designed the experiments, analyzed the data, prepared figures and/or tables, authored or reviewed drafts of the paper, and approved the final draft.
- Z. Jack Tseng conceived and designed the experiments, performed the experiments, analyzed the data, prepared figures and/or tables, authored or reviewed drafts of the paper, and approved the final draft.
- Shannon Brink conceived and designed the experiments, performed the experiments, analyzed the data, prepared figures and/or tables, authored or reviewed drafts of the paper, and approved the final draft.

## Data Availability

The raw measurements are available in the Supplemental File.

The videos and data for CT images of trials are available at Morphosource.

Video example of puncture experiments: https://doi.org/10.17602/M2/M159003 CT Scans of experimentally punctured cow humeri and radius-ulna: https://doi.org/10.17602/M2/M158996.

## Supplemental Information

Supplemental information for this article can be found online at http://dx.doi.org/10.7717/peerj.11450#supplemental-information.

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
