# Peer review of "Bite force estimates in juvenile Tyrannosaurus rex based on simulated puncture marks"

_PeerJ, doi:10.7717/peerj.11450_

## Round 0.1 · original submission · Minor Revisions

This is a concise, tightly written manuscript presenting a really interesting approach to tyrannosaurid paleobiology in particular and vertebrate feeding in general. The reviewers are quite positive on the manuscript also, and have presented some relatively minor suggestions for revision. I look forward to seeing the next version!

·

Basic reporting

Language is clear, and highly accessible: the authors even explain regression lines and confidence intervals in a caption. I suggest one additional reference (see "Validity of the findings).

The format can be improved slightly by distributing part of the discussion to the methods and results (lines 165-173). A brief take-home message about load rate sensitivity can be added to the discussion.

Experimental design

Excellent. It's unclear whether the printed cobalt model replicates denticles, which would presumably have an effect on the results.

Validity of the findings

Would denticles lower the necessary indentation forces? A discussion of the possibility is warranted.

Birds roll their heads vigorously with m. complexus and m. rectus capitis ventralis neck muscles, and broad-skulled T. rex would be notably good at it. A combination puppy dog tug-of-war and partial crocodilian death roll. Worth noting the possibility as a component of prey dismemberment by juvenile T. rex. A citation is suggested.

Additional comments

A refreshingly direct paper that will be influential for investigation of reptile feeding biomechanics. Interesting how Deinonychus and now juvenile T. rex show greater actualized bite forces than might be or have been expected from musculoskeletal simulations. It's intriguing to think about what we're missing in terms of muscle architecture and physiology.

Further comments and minor typo corrections are on the commented manuscript.

·

Basic reporting

The article is well written, with very clear use of the English language throughout. The literature in general is well rounded, however I have suggested some additional references in the "Validity of Findings" section. The article is well structured, however the abstract is missing at least a "Discussion" section.
From a different perspective, the link found on line 135 presents a page stating "No media found".
A minor comment: the acronym MDA in line 53 has already been defined on line 46

Experimental design

I am very interested in the present study, and would even be interested in replicating some of the experiments myself, however I have some small concerns about certain elements of the experimental analogies presented. While I am sure that they can easily be cleared up, I think it is important that some additional details be presented so as to ensure the most transparent publication of results.

(1) The authors need to clarify how they know the bite marks observed (lines 73 - 75) are without a doubt attributable to Tyrannosaurus. While they cite previous studies that argued these marks to be product of this specific predator, the present text should also confirm this, even if through a simple summary of the previous studies in 3 or 4 lines. I personally struggle to see how bite marks from this time period can be confirmed to be product of a specific animal, so think the text would benefit from clarifying this point.

(2) The authors need to clarify why they used a cobalt chromium alloy. Is this material truly analogous with the hardness of enamel? I am not a dentist, and I doubt many readers would be as well, however I have worked with tooth marks. If any scientist would be interested in replicating these experiments, then some clarification on why the authors used this material is necessary. From my understanding, metal implants are used in dentistry because they are less susceptible to corrosion, but to what point is the use of metal going to be conditioning the results?

(3) In light of the previous point, the authors need to clarify throughout the text (especially in the abstract) that the material used was an alloy. The abstract simply states that cobalt was used, which is misleading. The use of the term "dental grade" may also be useful for clearing up some confusion as well.

(4) Information about whether the bone used in experiments had meat intact or not is missing. When we work with bite marks, we have to consider that these traces are accidentally produced, few animals go around purposefully leaving bite marks on bone. Tooth marks are created when the tooth eventually comes into contact with bone, and more often than not this is product of the animal eating. If meat is intact, this will surely cushion the impact of tooth to bone and effect the results. If the tooth mark was created directly on the bone without anything in between, this will surely distort results and overestimate the true force. Did the authors take this into consideration? If not, I think this is an important point to take into consideration in future research.

(5) the authors report mean force values, however do not appear to take into consideration the best practices of descriptive statistics (at least as the text reads). If the measurements are not normally distributed, and present outliers, then mean values are not the most reliable metrics to report. The authors should first check the nature of the distribution (e.g. via a Shapiro-Wilks test), and upon determining whether the distribution is Gaussian or not, either report the mean (for Gaussian) or median (for non-Gaussian) values. I also ask the authors to please report some measurement of deviation, for example the standard deviation (for Gaussian) or Median Absolute Deviation (for non-Gaussian) values. These steps are quite simple to take, and can be performed in base R using the shapiro.test( ), mean( ), median( ), sd( ), and mad( ) functions.

Validity of the findings

The study is very interesting, and as stated in previous sections I would be very interested in looking into replicating these experiments with my own samples. The discussion of the paper presents a wide array of different observations that are welcoming and insightful.

My only "query" or comment is that I'm not sure about the use of the term "puncture" in some cases. Especially in the case of the marks observed on the vertebra. In taphonomic analyses of Pleistocene and some Holocene sites, I would argue that the marks found on the vertebra of Fig. 2 are actually pits, not punctures. However this may be due to my lack of knowledge on pre-Pleistocene tooth mark analyses. Nevertheless, the majority of terminology describes a puncture to be a mark where the tooth directly penetrates the cortical surface, leaving more of a hole than an indentation. When the tooth does not penetrate the surface, but instead simply leaves an indentation, this is referred to as a tooth pit. This observation is based on the terminology presented by the following authors;

Binford, L.R. (1981) Bones: Ancient Men and Modern Myths. New York: Academic Press Inc.

Haynes, G. (1983) A guide for differentiating mammalian carnivore taxa responsible for gnaw damage to herbivore limb bones, Paleobiology. 9(2):164-172

Fernández-Jalvo, Y.; Andrews, P. (2016) Atlas of Taphonomic Identifications. The Netherlands: Springer.

Additional comments

No further comments

·

Basic reporting

This manuscript calculates potential bite force in juvenile Tyrannosaurus rex by simulating known bite marks using mechanical indentation simulations. While bite force has been estimated across ontogeny within this group, previous studies have used alternate methods. The results presented here are slightly higher than the bite forces presented in the prior literature, and the authors frame their findings in terms of ecological niche partitioning across ontogenetic stages of the species.

The figures and table are all well-structured and presented, and all are critical to fully communicating the results of the study. The length is appropriate as well, and the quality of the writing is high. The only major omission of prior literature I noticed is probably related to its very recent publication. I think a discussion of its findings would strengthen the results:

Schroeder, Katlin, S. Kathleen Lyons, and Felisa A. Smith. "The influence of juvenile dinosaurs on community structure and diversity." Science 371.6532 (2021): 941-944.

(I also have a paper in press discussing similar eco-ontogenetic patterns in crocodyliforms coming out some time this month. I’d be happy to share it if the authors would like another point of comparison when discussing dietary niche transitions during growth and development.)

The general concept of the study is solid, with a few omissions I will discuss in more detail below. I think that a small amount of reorganization might provide clarity though. For example, the specific measurements of the tooth marks in the introduction (line 58) seem like details that should fall within the materials and methods. In the methods, only one speed of the indentations was mentioned (line 105), but in the discussion, multiple speeds are referenced (line 167). The varying speeds and why they were selection should be addressed in the methods. Similarly, the authors’ reasoning behind inclusion or exclusion of specific simulation runs is covered in the discussion (line 188), which seems like a subject that should be in the methods (such as near line 112). Many of the questions I had while reading would be cleared up with this slight restructuring.

Experimental design

This study largely follows the methods outlined in the Gignac et al (2010) paper. I did have a few questions about the experimental design that I think should be addressed in the text. Are there concerns that the structural integrity of the two bones were affected in later runs by breakage that occurred in earlier runs? Why were more bones/runs not used, especially in the higher speed indentations, whose n values are particularly low? Are the 17 runs presented in the Table the total number of simulations, or was there a first pass to remove runs where obvious movement of the specimen fouled the data? Could the authors include some kind of notation in Table 1 to show which were used in the total regression vs. the one that excluded incidences of breakage?

Also, in Figure 6, and the related sections of the results and discussion, what are the formulae of the regression lines? The results really need some metric of significance or goodness-of-fit of the regression line to the datapoints. The original paper by Gignac et al (2010) reports a linear formula and an R2 value for its regressions, both of which should be reported somewhere around line 120 and in the associated figure. The formula is mentioned in line 127, but I can’t find the actual results in the text. Also, starting on line 128, which of those results come from which of the two regressions’ formulae? Are they being averaged together? Perhaps reports both sets of predicted results and explicitly state which regression and formula were used in each.

In line 184, the authors are discussing some of their specific run results, but mention that the data are not shown. Why are they not shared? Perhaps exemplar graphs could be included, and then all of the runs appended to the manuscript for maximum transparency.

Validity of the findings

The results drawn from these simulations seem to align with previous estimates (though these are a little higher than those previously reported), but it is difficult to fully assess the validity of the results without the regression formulae and R2 values themselves.

Additional comments

I only a have a few, exceedingly minor comments that do not fit in the previous sections.

Does the company that provided the bones need to be named in the paper on line 97, or could they be placed in the acknowledgements? Also, I think that the hyphen needs to be removed from “in-tact”.

A quick skim through the citations revealed a couple of small typos: 1) Capitalize Eocene in line 308. Capitalize H in McHugh in line 327. The D in Deinonychus isn’t italicized in line 365.

---

## Round 0.2 · Minor Revisions

Thank you for your close attention to the comments from the reviewers. Overall, I think you have addressed all of them sufficiently. One final area where I would suggest a little more additional detail concerns the material used for the physical tooth model.

Reviewer 2 (Lloyd Courtenay) requested: "The authors need to clarify why they used a cobalt chromium alloy. Is this material truly analogous with the hardness of enamel? I am not a dentist, and I doubt many readers would be as well, however I have worked with tooth marks. If any scientist would be interested in replicating these experiments, then some clarification on why the authors used this material is necessary. From my understanding, metal implants are used in dentistry because they are less susceptible to corrosion, but to what point is the use of metal going to be conditioning the results?"

And you responded: "The Methods section was revised and reworded to explain the use of having the tooth reproduced in metal. It now reads: “In order to produce a suitable tooth analog, the STL file was 3D printed in a dental grade cobalt chromium alloy [Co(61.0),Cr(25.0),Mo(6.0),W(5.0),Mn(<1.0),Si(<1.0), Fe(<1.0)] with a yield strength of 47,436 N/cm2 (474.36 MPa) (Figure 3B) by the Argen corporation (San Diego, CA) to serve as a rigid model relative to the testing medium (i.e. cortical bone).”

The revised text is a good addition, but I feel that there should be just a little more to address the full reviewer comment. Presumably the metal *is* different in material properties from enamel/dentine--does this matter? Or not? Or is it just a necessary assumption? Was the most important thing that this be a rigid model, rather than strictly matching material properties? A brief statement explaining this should be included.

Thank you for your attention to this final comment!

---

## Round 0.3 · accepted · Accept

Thank you for your quick attention to the last round of comments!